# Point-of-care lung ultrasonography for early identification of mild COVID-19: a prospective cohort of outpatients in a Swiss screening center

Siméon Schaad [ID],[1] Thomas Brahier,[1] Mary-Anne Hartley,[2,3]
Jean-Baptiste Cordonnier,[3] Luca Bosso,[4] Tanguy Espejo,[4] Olivier Pantet,[5]
Olivier Hugli [ID],[4] Pierre-Nicolas Carron,[4] Jean-Yves Meuwly,[6]
Noémie Boillat-Blanco[1]

SS, TB and NB-B contributed equally.

For numbered affiliations see end of article.

**Correspondence to**
Dr Siméon Schaad;
simeon.schaad@unil.ch

## ABSTRACT

**Objectives** Early identification of SARS-CoV-2 infection is important to guide quarantine and reduce transmission. This study evaluates the diagnostic performance of lung ultrasound (LUS), an affordable, consumable-free point-of-care tool, for COVID-19 screening.

**Design, setting and participants** This prospective observational cohort included adults presenting with cough and/or dyspnoea at a SARS-CoV-2 screening centre of Lausanne University Hospital between 31 March and 8 May 2020.

**Interventions** Investigators recorded standardised LUS images and videos in 10 lung zones per patient. Two blinded independent experts reviewed LUS recording and classified abnormal findings according to prespecified criteria to investigate their predictive value to diagnose SARS-CoV-2 infection according to PCR on nasopharyngeal swabs (COVID-19 positive vs COVID-19 negative).

**Primary and secondary outcome measures** We finally combined LUS and clinical findings to derive a multivariate logistic regression diagnostic score.

**Results** Of 134 included patients, 23% (n=30/134) were COVID-19 positive and 77% (n=103/134) were COVID-19 negative; 85%, (n=114/134) cases were previously healthy healthcare workers presenting within 2–5 days of symptom onset (IQR). Abnormal LUS findings were significantly more frequent in COVID-19 positive compared with COVID-19 negative (45% vs 26%, p=0.045) and mostly consisted of focal pathologic B lines. Combining clinical findings in a multivariate logistic regression score had an area under the receiver operating curve of 80.3% to detect COVID-19, and slightly improved to 84.5% with the addition of LUS features.

**Conclusions** COVID-19-positive patients are significantly more likely to have lung pathology by LUS. However, LUS has an insufficient sensitivity and is not an appropriate screening tool in outpatients. LUS only adds little value to clinical features alone.

## INTRODUCTION

A year into the pandemic, COVID-19 remains a constant threat, overburdening the healthcare system. Current molecular diagnostic tests such as PCR and rapid antigen/antibody tests rely on consumables, which are vulnerable to shortages and saturation during exponential demand. The use of lung imaging as a diagnostic tool for COVID-19 has shown promises. CT scan of the chest has a good sensitivity for patients triaged in emergency departments[1 2] and has even been able to detect pathology in asymptomatic cases, suggesting its potential as an early screening test in specific populations.[3–5] However, CT scan and even X-rays expose patients to ionising radiation are costly, and often not available in decentralised screening sites. Lung ultrasonography (LUS) is an alternative, consumable-free, easy-to-use, portable, non-radiating and non-invasive screening tool that can be performed at the bedside, with simple disinfection between patients and only a negligible cost of ultrasound gel as a consumable. It would allow immediate identification of infected patients at the point-of-care and be invaluable to the sustainable control of the pandemic. Its diagnostic performance for pneumonia has been established using CT scan of the chest as a gold standard.[6] For COVID-19, recent studies conducted in emergency departments showed several LUS

patterns ranging from mild interstitial infiltrate to lung consolidation, which correlated with disease progression and outcome.[7 8] However, these studies included mostly severe patients in emergency departments or intensive care units, which may lead to overoptimistic diagnostic performance of LUS due to a spectrum effect.[9] To our knowledge, only one study included mild patients who did not need medical assessment, but the limited number of COVID-19-positive patients prevents us from drawing a conclusion.[10]

## STRENGTHS AND LIMITATIONS OF THIS STUDY

⇒ Acquisition and interpretation of lung ultrasound (LUS) images and videos were standardised using predefined patterns.
⇒ Ultrasound experts interpreted all LUS image and videos.
⇒ The study population consisted mainly of young and healthy healthcare workers, which prevents extrapolation of our results to an older and comorbid population.

This study aims to compare LUS characteristics between SARS-CoV-2 PCR-confirmed (COVID-19 positive) and PCR-negative (COVID-19 negative) patients in a screening centre and explore LUS performance for identification of COVID-19 outpatients.

## METHODS

### Study design, setting and population

This prospective cohort study recruited consecutive outpatients at the COVID-19 screening centre in Lausanne University Hospital, Switzerland (CHUV) between 31 March and 8 May 2020. All adults (age≥18 years) presenting at the centre with cough and/or dyspnoea and who fulfilled eligibility criteria for nasopharyngeal SARS-CoV-2 real-time (Rt-) PCR according to the state recommendations at the time of the study were eligible. These state criteria were the presence of symptoms suggestive of COVID-19 in a health worker or a patient with at least one vulnerability criterion, that is, age≥65 years old or having at least one comorbidity (obesity, diabetes, active cancer, chronic cardiovascular, pulmonary, liver, renal or inflammatory disease). Exclusion criteria were uninterpretable Rt-PCR results or absence of LUS recording. Written informed consent was obtained from all participants.

To ensure that LUS abnormal findings would be specific of a respiratory tract infection, we included a control group of healthy volunteers, matched for age (±5 years), sex and smoking status with COVID-19-positive patients (online supplemental table 1). These volunteers were asymptomatic during the previous 15 days (absence of odynophagia, cough, dyspnoea, runny nose, fever, loss of smell or taste) and did not have a documented SARS-CoV-2 infection.

At inclusion, demographics, comorbidities, symptoms (including duration) and vital signs were collected using a standardised electronic case report form in Research Electronic Data Capture. Patients were subsequently classified as either COVID-19 positive or COVID-19 negative according to the SARS-CoV-2 RT-PCR results (at inclusion or at any time during the 30-day follow-up if the test was repeated for the same clinical episode). We assessed 30-day outcome by phone using a standardised interview (persistence of symptoms, secondary medical consultation, hospital admission, death). The healthy controls were classified in a third group (healthy control group).

### Patient and public involvement

Patients were not involved in the design or conduct of this study.

### Sample size

The minimum sample size required for this study was 100 patients with a clinical suspicion of COVID-19. It was calculated using a COVID-19 prevalence of 20% and an estimated sensitivity of LUS to identify COVID-19 positive at 80%. This sample size guarantees a power of 80% with a false discovery rate of 5%.[11]

### Lung ultrasonography

Three medical students performed image acquisitions in the triage site. They were trained in LUS images acquisition with a 1-hour e-learning course and a 1-hour face-to-face practical course with an expert radiologist (J-YM). The first 10 acquisitions were done under direct supervision of an experienced board-certified expert (OP) who verified the quality of recorded images. Acquisition was standardised according to the '10-zone method',[12–14] consisting of 5 zones per hemithorax. Two images (sagittal and transverse) and 5 s videos were systematically recorded in every zone with a Butterfly IQ personal US system (Butterfly, Guiford, Connecticut, USA), using the lung preset. The LUS probe and the electronic tablet were disinfected with an alcohol-based solution between each patient to avoid nosocomial spread.[15]

For interpretation of LUS pathology, a physician experienced in LUS (TB) and an expert radiologist (J-YM), blinded to patients' diagnoses, independently filled a standardised report form as previously described.[8] The following patterns were reported for every zone: (1) normal appearance (A lines, <3 B lines), (2) pathologic B lines (≥3 B lines), (3) confluent B lines, (4) thickening of the pleura with pleural line irregularities (subpleural consolidation<1 cm) or (5) consolidation (≥1 cm). The presence of pleural effusion was also recorded.

Discordance between the two readers were adjudicated by a third expert (OP). The abnormal images were summed up in an LUS score for each patient, as previously described.[8 16 17]

### Statistical analyses

Differences between COVID-19-positive and COVID-19-negative patients for all collected demographic and clinical features as well as LUS findings and LUS score were evaluated by Mann-Whitney or $\chi^2$ test, as appropriate. A bilateral p value<0.05 was considered as indicative of statistical significance. A multivariate logistic regression was built from 22, 15, 10 and 8 features using recursive feature elimination (RFE), originally including the following:

(1) LUS findings (n=10).

▶ Number of zones with each of the five patterns (normal, pathological B lines, confluent B lines, pleural thickening, consolidation) (n=5).

- ► A dichotomised variable for the presence/absence of the above four pathological patterns detected (n=4).
- ► Binary variables for the presence of multifocal disease (n=1).
2) Symptoms at presentation (n=8).
- ► Binary variables for the presence of cough, sputum, dyspnoea, fever, anosmia, rhinorrhea, myalgia and diarrhoea.
3) Vital signs (n=3).
- ► Continuous variables for temperature, oxygen saturation and respiratory rate.
4) Epidemiological history (n=1)
- ► Binary variable for a history of known unprotected contact with a COVID-19 case.

Feature coefficients are presented, as well as their importance in ranked order from RFE. Performance at several stages of the RFE are reported, using the top 22, 15, 10 and 8 features. Models using just LUS or just clinical findings were also built.

Diagnostic performance is reported as sensitivity, specificity, positive and negative predictive values (PPV, NPV), positive and negative likelihood ratios (LR+, LR−) and area under the receiver operating curve (AUC). Due to the dataset size, we report findings on the entire dataset. A diagnostic score was derived from the summed coefficients, normalised within a range from −6 (COVID-19 positive highly unlikely) to +4 (COVID-19 positive highly likely) and the number of patients in each class are presented for each value of the score. The optimal cut-point was chosen using Youden index.[18]

The kappa coefficient was calculated to measure the inter-rater agreement between the two LUS readers. R Core Team (2019) statistical software and python V.3.0 with the sklearn library was used for analyses. Similar analyses were attempted on the outcome at 30-day follow-up but impossible due to the limited sample size.

The reporting of our results followed the STARD (Standards for Reporting Diagnostic accuracy studies) guidelines.

## RESULTS

### Demographics and clinical presentation
A total of 141 patients met inclusion criteria and were enrolled into the study; 7 (5%) were later excluded, due to uninterpretable PCR results or LUS technical issues (hospital's network connection problems). Of the 134 remaining patients, 31 (23%) were classified as COVID-19 positive and 103 (77%) as COVID-19 negative based on Rt-PCR test. Among the 13 COVID-19-negative patients who had a second screening test during the 30-day follow-up, only 1 had a positive SARS-CoV-2 Rt-PCR, related to a clearly distinct clinical episode. This patient was thus classified as COVID-19 negative. Most patients were female (63%), healthcare workers (85%) with a median age of 35 years; most sought out testing within the first 5 days of symptom onset (Table 1). COVID-19 positive patients had fewer comorbidities than COVID-19

negative, the latter suffering mostly from asthma, obesity or hypertension. COVID-19 positive patients presented more often with a history of fever and anosmia, but less often with dyspnoea than COVID-19-negative patients. Vital signs at inclusion were normal in most patients of both groups.

### Lung ultrasonography findings
Lung ultrasound was abnormal in 31% of patients (table 2). The two observers showed good concordance to differentiate a normal from an abnormal LUS, with a kappa of 0.67. Most anomalies were focal and unilateral. The most frequent patterns were pathologic B lines and thickening of the pleura with pleural line irregularities. Only 9.1% of control patients presented any abnormal finding on LUS, and all these anomalies were focal pathologic or confluent B lines (online supplemental tables 2 and 3).

Among all symptomatic patients, two factors were significantly associated with abnormal LUS: SARS-CoV-2 infection and history of fever (table 3). Indeed, COVID-19-positive patients had abnormal LUS findings significantly more frequently compared with COVID-19 negative (45% vs 26%, p=0.045). However, this feature alone was poorly sensitive (45%) and specific (74%). No specific ultrasonographic pattern on its own significantly distinguished COVID-19 positive from COVID-19-negative patients (table 2).

Although not statistically different, the proportion of COVID-19 positive with abnormal LUS findings was positively associated with symptoms duration. While only 30% of COVID-19-positive patients had abnormal LUS within 2 days of symptom onset, 52% of patients had pathological LUS after 2 days (p=0.24).

### Multivariate diagnostic score
We combined LUS findings with symptoms, vital signs and a binary feature for known contact with a COVID-19 case to build a multivariate logistic regression diagnostic score. Using all features, the score had 78.8% sensitivity, 84.0% specificity, 83.1% PPV, 61.4% NPV, 4.9 LR+, 0.3 LR− and 84.5% AUC (figure 1). We present a plot on which to assess the score according to a desired sensitivity/specificity trade-off.

In table 4, score performance with several combinations of features at various stages of RFE are presented. The strongest positive predictor was any evidence of pleural thickening at any number of sites (coefficient:+0.69) with LUS, although it became a negative predictor with an increasing number of sites with this feature (−0.40). The presence of pathological B lines and confluent pathological B lines was also positively associated with COVID-19 infection in this score. All three of the above patterns were retained by RFE within the top seven features. The LUS features that were negative and quickly eliminated by RFE were those describing consolidation and multifocal pathology. Cough, fever and anosmia were the highest ranked symptoms (coefficient≥0.4), in line with previous

**Table 1** Demographics, clinical characteristics and 30-day outcome of study participants according to nasopharyngeal Rt-PCR SARS-CoV-2 results

| | All (n=134) | SARS-Co-V2 positive (n=31) | SARS-CoV-2 negative (n=103) | P value |
|---|---|---|---|---|
| Demographics | | | | |
| Female sex | 84 (63) | 20 (65) | 64 (62) | 0.810 |
| Age, years; mean (SD) | 35.5 (29, 46) | 34 (26, 42) | 37 (29, 50) | 0.316 |
| Known contact with COVID-19 patient | 33 (28) | 10 (34) | 23 (25) | 0.334 |
| Current smoker | 39 (29) | 7 (23) | 32 (31) | 0.362 |
| Alcohol misuse | 3 (2.2) | 0 (0) | 3 (2.9) | 0.337 |
| Reason for testing | | | | |
| Vulnerable person* | 20 (15) | 6 (19) | 14 (14) | 0.430 |
| Healthcare worker | 114 (85) | 25 (81) | 89 (86) | 0.430 |
| Comorbidities | | | | |
| Any | 38 (28) | 3 (9.7) | 35 (34) | 0.008 |
| Hypertension | 10 (7.5) | 1 (3.2) | 9 (8.7) | 0.306 |
| Diabetes | 2 (1.) | 0 (0) | 2 (1.9) | 0.434 |
| Obesity | 16 (12) | 5 (16) | 11 (11) | 0.423 |
| Asthma | 17 (13) | 1 (3.2) | 16 (16) | 0.071 |
| Cardiovascular disease† | 5 (3.7) | 1 (3.2) | 4 (3.9) | 0.865 |
| Pulmonary disease‡ | 3 (2.2) | 0 (0) | 3 (2.9) | 0.337 |
| Active cancer | 3 (2.2) | 2 (6.5) | 1 (1.0) | 0.071 |
| Hepatitis or liver cirrhosis | 2 (1.4) | 0 (0) | 2 (1.9) | 0.434 |
| Chronic renal failure§ | 2 (1.4) | 0 (0) | 2 (1.9) | 0.434 |
| Chronic inflammatory disease | 4 (3.0) | 0 (0) | 4 (3.9) | 0.265 |
| Symptoms | | | | |
| Duration of symptoms, days; median (IQR) | 3 (2, 5) | 3 (2, 4) | 3 (2, 5) | 0.942 |
| Duration of symptoms | | | | 0.695 |
| 0–2 days | 50 (38) | 10 (32) | 40 (39) | |
| 3–5 days | 57 (43) | 18 (58) | 39 (38) | |
| ≥6 days | 26 (20) | 3 (9.7) | 23 (23) | |
| Cough | 118 (88) | 30 (97) | 88 (85) | 0.088 |
| Expectorations | 27 (20) | 10 (32) | 17 (17) | 0.055 |
| Dyspnoea | 79 (59) | 13 (42) | 66 (64) | 0.028 |
| History of fever | 75 (56) | 23 (74) | 52 (50) | 0.020 |
| Anosmia | 24 (18) | 10 (32) | 14 (14) | 0.017 |
| Rhinorrhea | 76 (57) | 20 (65) | 56 (54) | 0.317 |
| Odynophagia | 55 (41) | 13 (42) | 42 (41) | 0.908 |
| Myalgia | 91 (68) | 25 (81) | 66 (64) | 0.083 |
| Diarrhoea | 34 (25) | 5 (16) | 29 (28) | 0.177 |
| Temperature, °C; median (IQR) | 36.9 (36.6, 37.3) | 37 (36.7, 37.5) | 36.9 (36.6, 37.2) | 0.202 |
| Respiratory rate, beaths/minute; median (IQR) | 18 (16, 20) | 18 (14, 20) | 18 (16, 20) | 0.236 |
| Saturation, %; median (IQR) | 97 (97, 98) | 98 (97, 98) | 97 (97, 98) | 0.403 |
| Heart rate, beats/minute; median (IQR) | 86 (77, 95) | 87 (79, 90) | 86 (76, 98) | 0.955 |
| Follow-up at 30 days | | | | |
| Persistence of any symptoms at day 30 | 28 (23) | 12 (41) | 16 (17) | 0.008 |
| Fatigue | 14 (10) | 9 (29) | 5 (4.9) | 0.000 |

Continued

**Table 1** Continued

| | All (n=134) | SARS-Co-V2 positive (n=31) | SARS-CoV-2 negative (n=103) | P value |
|---|---|---|---|---|
| Myalgia | 6 (4.5) | 3 (9.7) | 3 (2.9) | 0.110 |
| Cough | 10 (7.4) | 3 (9.7) | 7 (6.8) | 0.592 |
| Expectoration | 2 (1.4) | 1 (3.2) | 1 (0.97) | 0.364 |
| Dyspnoea | 9 (6.7) | 6 (19) | 3 (2.9) | 0.001 |
| Fever | 2 (1.4) | 1 (3.2) | 1 (0.97) | 0.364 |
| Anosmia | 8 (6.0) | 7 (23) | 1 (0.97) | 0.000 |
| Rhinorrhea | 1 (0.8) | 1 (3.2) | 0 (0) | 0.067 |
| Odynodysphagia | 2 (1.4) | 1 (3.2) | 1 (0.97) | 0.364 |
| Diarrhoea | 2 (1.4) | 1 (3.2) | 1 (0.97) | 0.364 |
| Medical consultation during follow-up | 32 (26) | 9 (31) | 23 (25) | 0.521 |
| Hospitalisation/death | 0 (0) | 0 (0) | 0 (0) | |

Data are presented as n (%) unless indicated. Missing values: contact with infected people, 15; medical consultation at inclusion, 1; vital signs, 5; duration of symptoms, 1; obesity, 1.
*≥65 years old or comorbidity (obesity, diabetes, active cancer, chronic cardiovascular, pulmonary, liver, renal or inflammatory disease)
†Arrythmia, coronary disease.
‡Chronic obstructive pulmonary disease, fibrosis.
§Stage 3–5 according to chronic kidney disease classification.

reports. While LUS patterns were highly ranked in the RFE, rerunning the model without LUS findings reduced AUC by only 4% (AUC 84.5% vs 80.3%). LUS findings were poorly sensitive in the absence of clinical features (AUC: 63.9% sensitivity: 45.5%, specificity:77.3%, PPV: 66.7%, NPV: 55.6%, LR+: 2.0, LR–: 0.7).

Combining all 22 features and using RFE, we observe that removing 7 features had minimal impact on score performance, and removing 12 features reduces AUC by only 4% compared with the original.

### 30-day outcome

The 30-day follow-up was available for 121/134 (90%) patients. None was hospitalised or died during follow-up. COVID-19-positive patients had more frequently persistent symptoms (fatigue, dyspnoea or anosmia) at 30-day compared with COVID-19 negative (table 1).

The presence of an abnormal LUS at inclusion was not associated with symptom persistence (table 3).

As no patients were admitted or died, we could not analyse the value of LUS findings to predict critical clinical outcome.

### DISCUSSION

Lung pathology is detectable by CT scan of the chest early in the course of COVID-19 disease, even in asymptomatic patients, suggesting that lung imaging might have a place as a complementary diagnostic tool.[3] However, large-scale

**Table 2** Lung ultrasound characteristics of study participants according to nasopharyngeal Rt-PCR SARS-CoV-2 results

| | All (n=134) | SARS-CoV-2 positive (n=31) | SARS-CoV-2 negative (n=103) | P value |
|---|---|---|---|---|
| Abnormal lung ultrasound (any abnormal finding) | 41 (31) | 14 (45) | 27 (26) | 0.045 |
| Abnormal lung ultrasound, apart from focal B lines | 30 (22) | 11 (35) | 19 (18) | 0.046 |
| Multifocal | 16 (12) | 6 (19) | 10 (9.7) | 0.146 |
| Bilateral | 8 (6.0) | 3 (9.7) | 5 (4.9) | 0.320 |
| Number of pathologic zones; median (IQR) | 0 (0, 1) | 0 (0, 1) | 0 (0, 1) | 0.044 |
| Pathologic B lines (≥3) | 20 (15) | 6 (19) | 14 (14) | 0.430 |
| Confluent B lines (white lung) | 11 (8.2) | 4 (13) | 7 (6.8) | 0.277 |
| Pleural thickening | 18 (13) | 6 (19) | 12 (12) | 0.270 |
| Consolidations (>1 cm) | 1 (0.75) | 0 (0) | 1 (0.97) | 0.582 |
| Pleural effusion | 1 (0.75) | 0 (0) | 1 (0.97) | 0.000 |
| LUS score; median (IQR) | 0 (0, 1) | 0 (0, 3) | 0 (0, 1) | 0.044 |

**Table 3** Demographics and clinical characteristics of study participants according to the presence of an abnormal lung ultrasound

| | All (n=134) | Abnormal LUS (n=41) | Normal LUS (n=93) | P value |
|---|---|---|---|---|
| Demographics | | | | |
| Female sex | 84 (63) | 28 (68) | 56 (60) | 0.373 |
| Age; median (IQR) | 35.5 (29, 46) | 38 (31, 48) | 35 (28, 45) | 0.574 |
| Current cigarette smoker | 39 (29) | 12 (29) | 27 (29) | 0.978 |
| Alcohol misuse | 3 (2.2) | 0 (0) | 3 (3.2) | 0.245 |
| Reason of testing | | | | |
| Vulnerable person§ | 20 (15) | 3 (7.3) | 17 (18) | 0.101 |
| Healthcare worker | 114 (85) | 38 (93) | 76 (82) | 0.101 |
| Positive Rt-PCR result | 31 (23) | 14 (34) | 17 (18) | 0.045 |
| Comorbidities | | | | |
| Any | 38 (28) | 13 (32) | 25 (27) | 0.568 |
| Hypertension | 10 (7.5) | 3 (7.3) | 7 (7.5) | 0.966 |
| Diabetes | 2 (1.5) | 1 (2.4) | 1 (1.1) | 0.549 |
| Obesity | 16 (12) | 3 (7.3) | 13 (14) | 0.265 |
| Asthma | 17 (13) | 7 (17) | 10 (11) | 0.311 |
| Cardiovascular disease* | 5 (3.7) | 2 (4.9) | 3 (3.2) | 0.642 |
| Pulmonary disease† | 3 (2.2) | 0 (0) | 3 (3.2) | 0.245 |
| Active cancer | 3 (2.2) | 1 (2.4) | 2 (2.2) | 0.917 |
| Hepatitis or liver cirrhosis | 2 (1.5) | 1 (2.4) | 1 (1.1) | 0.549 |
| Chronic renal failure‡ | 2 (1.5) | 0 (0) | 2 (2.2) | 0.344 |
| Chronic inflammatory disease | 4 (3.0) | 0 (0) | 4 (4.3) | 0.178 |
| Symptoms | | | | |
| Duration of symptoms, days; median (IQR) | 3 (2, 5) | 3 (2, 5) | 3 (2, 5) | 0.344 |
| Duration of symptoms | | | | 0.210 |
| 0–2 days | 50 (38) | 11 (22) | 39 (78) | |
| 3–5 days | 57 (43) | 21 (37) | 36 (63) | |
| ≥6 days | 26 (20) | 9 (35) | 17 (65) | |
| Cough | 118 (88) | 34 (83) | 84 (90) | 0.224 |
| Expectorations | 27 (20) | 7 (17) | 20 (22) | 0.556 |
| Dyspnoea | 79 (59) | 25 (61) | 54 (58) | 0.752 |
| Hemoptysis | 2 (1.5) | 0 (0) | 2 (2.2) | 0.344 |
| History of fever | 75 (56) | 29 (71) | 46 (49) | 0.022 |
| Anosmia | 24 (18) | 11 (27) | 13 (14) | 0.074 |
| Rhinorrhea | 76 (57) | 21 (51) | 55 (59) | 0.394 |
| Odynophagia | 55 (41) | 17 (41) | 38 (41) | 0.948 |
| Myalgia | 91 (68) | 31 (76) | 60 (65) | 0.205 |
| Diarrhoea | 34 (25) | 8 (20) | 26 (28) | 0.301 |
| Temperature, °C; median (IQR) | 36.9 (36.6, 37.3) | 37 (36.6, 37.5) | 36.9 (36.6, 37.2) | 0.270 |
| Respiratory rate, beaths/minute; median (IQR) | 18 (16, 20) | 18 (16, 20) | 18 (16, 20) | 0.330 |
| Saturation, %; median (IQR) | 97 (97, 98) | 97 (97, 98) | 97 (97, 98) | 0.385 |
| Heart rate, beats/minute; median (IQR) | 86 (77, 95) | 88 (79, 98) | 85 (76.5, 94) | 0.170 |
| Follow-up at 30 days | | | | |
| Persistence of any symptoms at day 30 | 28 (23) | 9 (24) | 19 (23) | 0.924 |
| Fatigue | 14 (10) | 7 (17) | 7 (7.5) | 0.096 |

**Table 3** Continued

|  | All (n=134) | Abnormal LUS (n=41) | Normal LUS (n=93) | P value |
|---|---|---|---|---|
| Myalgia | 6 (4.5) | 2 (4.9) | 4 (4.3) | 0.882 |
| Cough | 10 (7.5) | 3 (7.3) | 7 (7.5) | 0.966 |
| Expectorations | 2 (1.5) | 0 (0) | 2 (2.2) | 0.344 |
| Dyspnoea | 9 (6.7) | 4 (9.8) | 5 (5.4) | 0.351 |
| Fever | 2 (1.5) | 0 (0) | 2 (2.2) | 0.344 |
| Anosmia | 8 (6.0) | 1 (2.4) | 7 (7.5) | 0.252 |
| Rhinorrhea | 1 (.75) | 0 (0) | 1 (1.1) | 0.505 |
| Odynophagia | 2 (1.5) | 1 (2.4) | 1 (1.1) | 0.549 |
| Diarrhoea | 2 (1.5) | 0 (0) | 2 (2.2) | 0.344 |
| Medical consultation during follow-up | 26 (21) | 10 (26) | 16 (19) | 0.364 |
| Hospitalisation/death | 0 (0) | 0 (0) | 0 (0) |  |

Data are presented as n (%) unless otherwise indicated.
*Arrythmia, coronary disease.
†Chronic obstructive pulmonary disease, fibrosis.
‡Stage 3–5 according to chronic kidney disease classification.
§≥ 65 years old or comorbidity (obesity, diabetes, active cancer, chronic cardiovascular, pulmonary, liver, renal or inflammatory disease)
Rt, real-time.

CT screening is not feasible even in hospital settings with abundant resources. Point-of-care LUS is now affordable, portable and implementable in a decentralised setting and has all the attributes to become a pragmatic community-based screening tool.

We evaluated the diagnostic performance of LUS in a prospective cohort of patients with mild acute respiratory tract infection attending a COVID-19 Swiss screening centre. COVID-19 positive outpatients more frequently had abnormal LUS findings at inclusion compared with COVID-19 negative. However, LUS findings alone had insufficient sensitivity, NPV and LR– to recommend LUS as an independent screening tool in outpatients. The combination of both LUS and clinical features in a multivariate regression score showed that LUS features only adds little value to clinical features alone regarding the prediction of COVID-19.

The limited sensitivity of LUS in our population is discordant with previous studies, which showed a sensitivity varying from 62% to 97% to identify Rt-PCR-confirmed COVID-19. These retrospective studies were conducted in emergency departments and included patients with severe and critical COVID-19 infection.[19–21] Some studies included mild patients who were evaluated in the ED and sometimes hospitalised.[22–24] Although these patients had mild COVID-19, their disease was more severe as they needed a medical assessment unlike the patients included in the present study who came for SARS-CoV-2 screening.

Other studies using CT scan of the chest also showed an excellent sensitivity (97%–98%) to diagnose

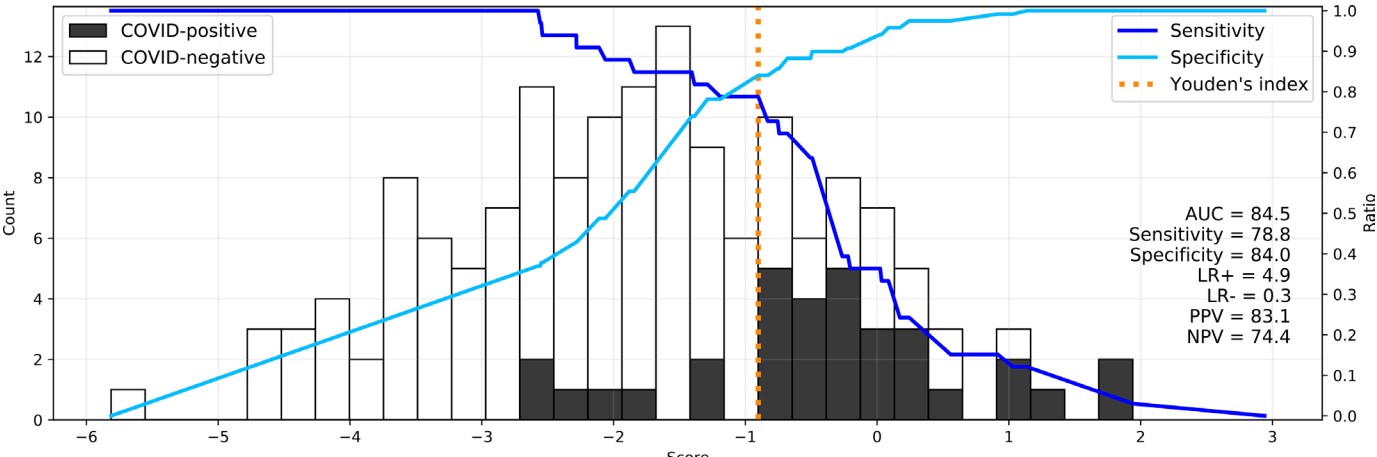

**Figure 1** A multivariate logistic regression diagnostic score (x axis) to discriminate COVID-19 positive from COVID-19-negative patients (black and white bars, respectively, with count on y axis). Sensitivity (—) and specificity (—) of the score are plotted with Youden's index (sensitivity+specificity−1) marked in orange. All 22 features are used in the depicted image on a model trained on all data points.

**Table 4** Multivariate logistic regression for COVID-19 diagnosis

| RFE selection order | Feature groups — LUS findings (n=10) | Symptoms (n=8) | Vital signs (n=3) | Epidemiological history (n=1) | Coefficient* Neg | Coefficient* Pos | Diagnostic performance with various feature sets — 22–0 features=22 / 10 LUS / 8 symptoms / 1 contact / 3 signs | 22–7 features=15 / 6 LUS / 8 symptoms / 1 contact / No signs | 22–12 features=10 / 5 LUS / 5 symptoms / 1 contact / No signs | 22–14 features=8 / 5 LUS / 3 symptoms / No contact / No signs |
|---|---|---|---|---|---|---|---|---|---|---|
| 1 (removed last) | | Cough | | | | 0.40 | Sens: 78.8% | Sens: 75.8% | Sens: 84.8% | Sens: 81.8% |
| 2 | Pleural thickening (any) | | | | | 0.69 | Spec: 84.0% | Spec:83.2% | Spec: 72.3% | Spec: 62.2% |
| 3 | Pleural thickening (number of sites) | | | | −0.40 | | AUC: 84.5% | AUC: 83.5% | AUC: 80.2% | AUC: 76.6% |
| 4 | | Fever | | | | 0.44 | LR+: 4.9 | LR+: 4.5 | LR+: 3.1 | LR+: 2.2 |
| 5 | Confluent B lines (number of sites) | | | | | 0.41 | LR−: 0.3 | LR−: 0.3 | LR−: 0.2 | LR−: 0.3 |
| 6 | Normal pattern (number of sites) | | | | | 0.29 | PPV: 83.1% | PPV: 81.8% | PPV: 75.4% | PPV: 68.4% |
| 7 | Pathologic B lines (number of sites) | | | | | 0.49 | NPV: 61.4% | NPV: 80.6% | NPV: 73.5% | PPV: 64.7% |
| 8 | | Anosmia | | | | 0.43 | | | | |
| 9 | | | | Contact with COVID-19 | | 0.47 | | | | |
| 10 | | Dyspnoea | | | −0.28 | | | | | |
| 11 | | Myalgia | | | | 0.37 | | | | |
| 12 | | Diarrhoea | | | −0.49 | | | | | |
| 13 | Multifocality | | | | −0.26 | | | | | |
| 14 | | Rhinorrhea | | | | 0.35 | | | | |
| 15 | | Sputum | | | | 0.41 | | | LUS findings only | Clinical only |
| 16 | | | Oxygen saturation | | | 0.20 | | | Sens: 45.5% | Sens: 72.7% |
| 17 | Consolidation (any) | | | | −0.18 | | | | Spec: 77.3% | Spec: 79.8% |
| 18 | | | Temperature (°C) | | | 0.22 | | | AUC: 63.9% | AUC: 80.3% |
| 19 | | | Respiratory rate | | −0.30 | | | | LR+: 2.0 | LR+: 3.6 |
| 20 | Consolidation (any) | | | | −0.18 | | | | LR−: 0.7 | LR−: 0.3 |
| 21 | Pathologic B lines (any) | | | | −0.07 | | | | PPV: 66.7% | PPV: 78.3% |
| 22 (removed first) | Confluent B lines (any) | | | | | 0.26 | | | NPV: 55.6% | NPV: 64.5% |

Multivariate logistic regression for COVID-19 diagnosis where selection order is indirectly proportional to the feature's predictive importance, in RFE, that is, the feature labeled 22 was removed first, while 1 was retained until the end. Four feature groups containing 10 LUS findings, 8 symptoms, 3 vital signs and 1 epidemiological history of contact are color coded according to their coefficient in the multivariate score including all 22 features (Pos is positive correlation with COVID-19 and Neg is negative correlation).

*The coefficient in multivariate scores is susceptible to multicollinearity.

LR+, positive likelihood ratio; LR−, negative likelihood ratio; NPV, negative predictive value; PPV, positive predictive value; RFE, recursive feature elimination.

COVID-19.[2 25 26] However, all these studies were conducted in hospitalised patients, preventing extrapolation to our milder population screened for symptoms only.

The clinical severity of the disease strongly affects the performance of diagnostic tests, and particularly the sensitivity of LUS. We conclude that while LUS may be an interesting COVID-19 screening tool in emergency departments, it is not reliable when used alone in patients with mild disease. In the only study investigating CT scan of the chest features in patients with asymptomatic (73%) or mild (27%) COVID-19, which was conducted in the passengers of the cruise ship Diamond Princess, 54% of asymptomatic patients and 79% of patients with mild disease presented opacities on CT scan of the chest. These results suggested the potential use of CT scan of the chest in clinical decision-making.[3] Most opacities were located in the peripheral areas of the lung, where LUS is performant. Patients included in the Diamond Princess Study were older compared with our study population (mean of 63±15 years vs 39±13 years), a possible explanation for the lower proportion of patients with lung involvement in our study.

Another potential explanation of the discrepancy between our study and previous publications is the short duration of symptoms at presentation. Although we did not confirm this association with our data, a previous study described a relationship between the duration of infection and the proportion of abnormal radiological findings.[27–29] In one study, only 44% of patients presenting within 2 days of symptoms had an abnormal CT, while this proportion rose to 91% after 3–5 days and 96% after 5 days.[29] This study did not provide any data on COVID-19 severity. In another study using X-ray of the chest in patients admitted to the emergency department, the proportion of an abnormal X-ray of the chest increased with the duration of symptoms (63% in the first 2 days to 84% after 9 days).[30] In our study, we did not confirm this hypothesis; however, we observed more abnormal LUS findings in COVID-19-positive patients who had more than 2 days of symptoms (52% vs 30%), although our results were not statistically significant.

In our study, most patients with abnormal LUS findings presented with focal pathologic B lines, confluent B lines or pleural thickening, irrespective of the aetiology of the acute respiratory tract infection. Inclusion of healthy volunteers confirmed the causality between LUS findings and acute respiratory tract infections. Indeed, only 9% of healthy volunteers presented LUS anomalies (and all were focal pathologic B lines).

Two previous studies showed that thickened pleural lines on LUS were significantly associated with COVID-19.[19 20] However, in a third report, LUS findings were similar in both COVID-19 and non-COVID-19 patients.[21]

## LIMITATIONS

Our study has some limitations. First, most of our patients were healthy and young healthcare workers, which prevents extrapolation of our results to an older and comorbid population. However, young, healthy patients are of a prime importance in the management of the virus spread.[31] Second, SARS-CoV-2 Rt-PCR nasopharyngeal swab was used as the gold standard, and we might have missed some early infections when it has limited sensitivity.[32] However, it is considered as the reference diagnostic method. Furthermore, we sought to mitigate technical and sample collection error using validated nucleic acid amplification tests and a dedicated trained medical team performing nasopharyngeal swabs.[33] In addition, we had 30-day follow-up, which may have reduced the number of patients misclassified as COVID-19 negative. Third, medical students, and not ultrasound experts, performed LUS images and videos acquisition. However, they had a focused training by experts and followed a standardised image acquisition protocol. To better investigate the predictive potential of LUS findings, we built a multivariate score. The small sample size and high feature count (n=22) exposes the model to the risk of overfitting. Thus, this score is not ready for clinical use, but rather is a mean to demonstrate the feature importance by RFE.

## CONCLUSION

To our knowledge, this is the first study, which assessed the use of LUS in a screening centre outpatient population with mild COVID-19. As disease severity plays an important role in the ultrasonographic findings, LUS is poorly sensitive as a SARS-CoV-2 screening tool in the context of mild community-level screening.

**Author affiliations**
[1]Infectious Diseases Service, University Hospital of Lausanne, Lausanne, Switzerland
[2]Digital global Health Department, University of Lausanne, Lausanne, Switzerland
[3]Machine Learning and Optimization Laboratory, EPFL, Lausanne, Switzerland
[4]Emergency Department, Lausanne University Hospital Emergency Care Service, Lausanne, Switzerland
[5]Adult Intensive Care Unit, Lausanne University Hospital, Lausanne, Switzerland
[6]Department of Radiology, Lausanne University Hospital Division of Radiodiagnostics and Interventional Radiology, Lausanne, Switzerland

**Correction notice** This article has been corrected since it was first published. The licence type has been updated to CC BY.

**Acknowledgements** We thank all the patients who accepted to participate and make this study possible. We thank all healthcare workers of the triage unit of the emergency department of the University Hospital of Lausanne who supported the study and managed COVID-19-suspected patients.

**Contributors** J-YM, OH, P-NC and NB-B: study conception, study design, study performance, study management, data analysis, data interpretation and manuscript writing. SS, TB, J-YM and OP: LUS images review, data interpretation and critical review of the manuscript. TE and LB: LUS images recording, data interpretation and critical review of the manuscript. M-AH and J-BC: data analysis, interpretation, visualisations and critical review of the manuscript. All authors approved the final version of the manuscript and agreed to be accountable for all aspects of the work in ensuring that questions related to the accuracy or integrity of any part of the work are appropriately investigated and resolved. NB-B had full access to all the data in the study and takes responsibility for the integrity of the data and the accuracy of the data analysis. NBB accepts full responsibility for the work and/ or the conduct of the study, had access to the data, and controlled the decision to publish

**Funding** This work was supported by an academic award of the Leenaards Foundation (to NB-B), by the Foundation of Lausanne University Hospital, and the Emergency Department Lausanne University Hospital. The funding bodies had no role in the design of the study and collection, analysis and interpretation of data and in writing the manuscript.

**Competing interests** None declared.

**Patient and public involvement** Patients and/or the public were not involved in the design, or conduct, or reporting, or dissemination plans of this research.

**Patient consent for publication** Not applicable.

**Ethics approval** This study involves human participants and was approved by Swiss Ethics Committee of the canton of Vaud (CER-VD 2019-02283). Participants gave informed consent to participate in the study before taking part.

**Provenance and peer review** Not commissioned; externally peer reviewed.

**Data availability statement** Data are available in a public, open access repository. Dataset available from https://zenodo.org/record/4617904#.Ya-gfi3pOu6.

**ORCID iDs**
Siméon Schaad http://orcid.org/0000-0003-3110-071X
Olivier Hugli http://orcid.org/0000-0003-2312-1625

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
