## [Reviewer comments · BMJ Open]

ARTICLE DETAILS

TITLE (PROVISIONAL)	Point-of-care lung ultrasonography for early identification of mild COVID-19: a prospective cohort of outpatients in a Swiss screening center
AUTHORS	Schaad, Siméon; Brahier, Thomas; HARTLEY, Mary-Anne; CORDONNIER, Jean-Baptiste; BOSSO, Luca; ESPEJO, Tanguy; PANTET, Olivier; Hugli, Olivier; Carron, Pierre-Nicolas; MEUWLY, Jean-Yves; Boillat-Blanco, Noémie

VERSION 1 – REVIEW

REVIEWER	Shirley Friedman Tel Aviv Sourasky Medical Center
REVIEW RETURNED	23-Jan-2022

GENERAL COMMENTS	Dear Authors, I enjoyed reading your well-written paper on the reliability of point of care LUS for early identification of mild COVID-19. The paper indeed highlights the use of LUS in outpatient clinics with LRTI symptoms as opposed to the majority of the published data exploring LUS in ED and hospitalized patients. Having medical students record the LUS further emphasizes the feasibility of teaching and performing point of care LUS in various medical settings with different resources. The conclusion that LUS by itself is of small value to the clinical diagnosis of covid 19 in this cohort is not surprising. LUS is a "piece of a puzzle" and in healthy outpatients with LRTI symptoms, the predictive value of specific covid symptoms such as anosmia is expected to outweigh the LUS findings, which may be present in other viral LRTI's. I have two minor comments that I believe will further clarify certain methodological aspects and strengthen the paper. 1. In the statistical analysis section of the methods (page 7 line 148) it is written "Number of pathological zones for each of the five patterns (normal, pathological B lines, confluent B lines, pleural thickening, consolidation) (n=5)" As "normal" is not a pathological pattern I would suggest changing the phrasing of that sentence. 2. The addition of a control group composed of healthy volunteers is indeed important. The volunteers were matched to the covid positive patients. The analysis as presented in supplementary table 1 compares the LUS findings between the healthy control and the entire LRTI cohort.
---

	It is my opinion that adding an analysis comparing the healthy controls to the covid positive patient to which they were matched and further elaboration of this issue will be of benefit to the value of the study. Thank you for the opportunity to review this interesting work.
--	---

REVIEWER	Laurent Zieleskiewicz Aix-Marseille Université, Department of Anesthesiology and Intensive Care Medicine, Hôpital Nord, Assistance Publique Hôpitaux de Marseille
REVIEW RETURNED	01-Feb-2022

GENERAL COMMENTS	This is a prospective observational study evaluating the diagnostic performance of lung ultrasound for mild COVID-19. The authors are experts in the field as they recently published a study in this area (are there patients in common in both studies?). My main concern is the utility of this study. How could the LUS be used as the sole diagnostic screening tool, especially in asymptomatic patients? Clearly, many patients will be misclassified as negative due to the absence of pneumonia. Therefore, the LUS cannot be used as a diagnostic screening tool to assign isolation. The authors state that "no study has described the results of LUS in subjects with mild COVID-19." "This is not entirely correct because several studies have included very mild patients. We can cite the Speidel study PMID: 33487473 in which the patients' PAO2 was 65 in room air. In the Lichter study, PMID: 32860069, the mean O2 saturation in room air was 95%. In the Volpicelli study, 400 patients were in the mild phenotype (no or very few clinical signs) PMID: 33743018. Finally, a similar study (but of lower quality with surprising results) was recently published: Lung Ultrasonography in Ruling Out COVID-19 Among Health Care Workers in Two Italian Emergency Departments: A Multicenter Study https://doi.org/10.1177/87564793211037607. That said, the study is well written and presented as a negative study, as it should be. Minor Concern: Abstract OK Introduction OK except for the fact that no study was conducted in mild patients (ditto in the discussion section). Methods: Resident training time for LUS is relatively short (10 exams rather than 25). https://doi.org/10.1097/ALN.0000000000003096 Discussion: "Consistent with our findings, a relationship between duration of infection and proportion of abnormal radiological findings has been described [22-24]." No you did not find this result, it was not significant.
---

VERSION 1 – AUTHOR RESPONSE

Reviewer #1:

1. In the statistical analysis section of the methods (page 7 line 148) it is written "Number of pathological zones for each of the five patterns (normal, pathological B lines, confluent B lines, pleural thickening, consolidation) (n=5)"

We agree with the reviewer that this sentence was misleading. We changed the manuscript accordingly: "Number of zones with each of the five patterns (normal, pathological B lines, confluent B lines, pleural thickening, consolidation)."

2. The addition of a control group composed of healthy volunteers is indeed important. The volunteers were matched to the covid positive patients. The analysis as presented in supplementary table 1 compares the LUS findings between the healthy control and the entire LRTI cohort. It is my opinion that adding an analysis comparing the healthy controls to the covid positive patient to which they were matched and further elaboration of this issue will be of benefit to the value of the study.

We added a supplementary table 3 showing the comparison between ultrasound findings in covid positive patients and healthy volunteers. We get similar results when comparing covid-positive patients with healthy controls than when comparing the entire LRTI cohort with healthy controls.

Reviewer #2

The authors are experts in the field as they recently published a study in this area (are there patients in common in both studies?).

No patients are included in both studies as the studies were conducted in different settings. The present study was conducted in a screening center where patients did not have a medical assessment. The study of Brahier et al (<https://doi.org/10.1093/cid/ciaa1408>) was conducted in the emergency department of the hospital and the majority of these patients (79%) were hospitalized. The severity of the disease was not comparable between these two studies.

The authors state that "no study has described the results of LUS in subjects with mild COVID-19." "This is not entirely correct because several studies have included very mild patients. We can cite the Speidel study PMID: 33487473 in which the patients' PAO2 was 65 in room air. In the Lichter study, PMID: 32860069, the mean O2 saturation in room air was 95%. In the Volpicelli study, 400 patients were in the mild phenotype (no or very few clinical signs) PMID: 33743018. Finally, a similar study (but of lower quality with surprising results) was recently published: Lung Ultrasonography in Ruling Out COVID-19 Among Health Care Workers in Two Italian Emergency Departments: A Multicenter Study

We thank the reviewer for these references. We agree that patients included in the Speidel, Lichter and Volpicelli studies have mild COVID-19 according to the WHO classification. However, these patients attended the emergency department and/or were hospitalized (in Speidel and Lichter studies), which indicate more severe symptoms. Our study was conducted in a screening center, with no medical assessment whatsoever.

The patients' population of the Copetti study is similar to ours. However, the limited number of SARS-CoV-2 subjects (N=2) prevents us from drawing a conclusion.

We added the reference of this study in the introduction and discussion sections of the manuscript and highlighted the differences between mild COVID-19 patients who needed a medical assessment and our study population who came for SARS-CoV-2 screening only.

"Consistent with our findings, a relationship between duration of infection and proportion of abnormal radiological findings has been described [22-24]." No you did not find this result, it was not significant.

We agree with the reviewer and changed the manuscript accordingly.

Methods: Resident training time for LUS is relatively short (10 exams rather than 25).

A study showed that after scanning 11 zones, novice learners are able to achieve proficiency for quantifying B-lines on LUS. We added this reference to support our training (<https://doi.org/10.1002/ehf2.12907>).

We also would like to emphasize that the role of medical students was to record LUS images and videos following a standard procedure, and not to interpret them. As mentioned in the methods section, two to three LUS experts interpreted the images and videos retrospectively.

VERSION 2 – REVIEW

REVIEWER	Shirley Friedman Tel Aviv Sourasky Medical Center
REVIEW RETURNED	14-Apr-2022

GENERAL COMMENTS	Thank you for the opportunity to review the revised manuscript. My remarks were addressed appropriately. I think the discussion would benefit from mentioning that LUS COVID findings are not specific and may be present in other viral LRTI's as was demonstrated in table 2. This also contributes to the conclusion that "LUS has a insufficient sensitivity and is not an appropriate screening tool in outpatients. LUS only adds little value to clinical features alone" as LUS may be abnormal in any other viral infection.
---

REVIEWER	Laurent Zieleskiewicz Aix-Marseille Université, Department of Anesthesiology and Intensive Care Medicine, Hôpital Nord, Assistance Publique Hôpitaux de Marseille
REVIEW RETURNED	04-Apr-2022

GENERAL COMMENTS	Authors should be congratulated for their work. They adressed very clearly all my comments.
---